# Cytogenetic Damage Induced by Radioiodine Therapy: A Follow-Up Case Study

**DOI:** 10.3390/ijms24065128

**Published:** 2023-03-07

**Authors:** Igor K. Khvostunov, Elena Nasonova, Valeriy Krylov, Andrei Rodichev, Tatiana Kochetova, Natalia Shepel, Olga Korovchuk, Polina Kutsalo, Petr Shegai, Andrei Kaprin

**Affiliations:** 1A.F. Tsyb Medical Radiological Research Center (MRRC)—Branch of the National Medical Research Radiological Center of the Ministry of Health of the Russian Federation, 4 Koroliova St., 249036 Obninsk, Russia; 2Joint Institute for Nuclear Research (JINR), 6 Joliot-Curie St., 141980 Dubna, Russia; 3Federal State Budgetary Institution, National Medical Research Radiological Center of the Ministry of Health of the Russian Federation, 2 Botkinskiy Proezd, 125284 Moscow, Russia; 4Federal State Autonomous Educational Institution of Higher Professional Education, Department of Oncology and Radiology Named after N.P. Kharchenko, Medical Institute, Peoples’ Friendship University of Russia, 117198 Moscow, Russia

**Keywords:** thyroid cancer, radioiodine therapy, side effect, radiation marker, cytogenetics, biodosimetry, chromosomal aberrations, blood lymphocytes, mFISH

## Abstract

The risk of toxicity attributable to radioiodine therapy (RIT) remains a subject of ongoing research, with a whole-body dose of 2 Gy proposed as a safe limit. This article evaluates the RIT-induced cytogenetic damage in two rare differentiated thyroid cancer (DTC) cases, including the first follow-up study of a pediatric DTC patient. Chromosome damage in the patient’s peripheral blood lymphocytes (PBL) was examined using conventional metaphase assay, painting of chromosomes 2, 4, and 12 (FISH), and multiplex fluorescence in situ hybridization (mFISH). Patient 1 (female, 1.6 y.o.) received four RIT courses over 1.1 years. Patient 2 (female, 49 y.o.) received 12 courses over 6.4 years, the last two of which were examined. Blood samples were collected before and 3–4 days after the treatment. Chromosome aberrations (CA) analyzed by conventional and FISH methods were converted to a whole-body dose accounting for the dose rate effect. The mFISH method showed an increase in total aberrant cell frequency following each RIT course, while cells carrying unstable aberrations predominated in the yield. The proportion of cells containing stable CA associated with long-term cytogenetic risk remained mostly unchanged during follow-up for both patients. A one-time administration of RIT was safe, as the threshold of 2 Gy for the whole-body dose was not exceeded. The risk of side effects projected from RIT-attributable cytogenetic damage was low, suggesting a good long-term prognosis. In rare cases, such as the ones reviewed in this study, individual planning based on cytogenetic biodosimetry is strongly recommended.

## 1. Introduction

RIT is considered the “gold standard” for DTC treatment as the only known cure for distant metastases [1]. While RIT is used to target pathological areas, radiation exposure of healthy tissues still occurs, and the risk-to-benefit ratio remains a subject of ongoing investigations [2,3]. No severe hematological side effects have been observed in patients whose whole-body dose was 2 Gy or less. Therefore, this empirically established value has been used as a benchmark for the dose constraint [4]. Since the biometrics of DTC patients vary significantly, accurate methods for assessing the exposure of healthy tissues are necessary [5]. This is particularly important for distinctive DTC patients, whose treatment requires individual planning [6,7]. Biodosimetry based on CA in PBL has been proposed as an efficient solution to the problem, as certain types of CA are radiation markers and have been used to estimate dose to healthy tissues in DTC patients [8,9,10].

RIT has been associated with an increased risk of secondary malignancy—mainly leukemia—in some studies [11,12]. However, a meta-analysis of the published data on the application of RIT for thyroid cancer revealed an overall low probability of secondary leukemia [2]. Additional investigations are needed to resolve this ambiguity. Further study of residual damage in PBL is also important for prognostic purposes [13]. 

Since DTC is rarely seen in children and adolescents, there are no well-established guidelines for treating children with RIT. This study reviews the results of the first-ever follow-up examination of the rare case of Patient 1 (Pat1), who received four RIT courses over the period of 1.1 years (1.6 through 2.6 years old). Rare DTC cases also include patients with a very high total administered radioiodine activity. Patient 2 (Pat2) (female, 49) received 10 RIT courses over the period of 5.4 years before her first examination that was reviewed in this study. In total, 12 RIT courses were administered over a period of 6.4 years. With a cumulative administered activity of ^131^I exceeding approximately 39 GBq, a reliable substantiation for RIT extension was required.

The goal of this follow-up study of two rare DTC cases was to investigate the RIT-induced CA by using three cytogenetic assays [14], including the most advanced, highly sensitive molecular method of mFISH [15]. This method was developed to probe the entire genome and thus provides additional information for risk estimation and prognosis.

## 2. Results

Table 1 shows CAs observed in the PBL of the DTC patients and healthy donors. For Pat1, day 0 means the date of initial blood sampling; days 135, 281, and 400—sampling before the subsequent courses of RIT; and days 4, 138, 285, and 404—sampling after RIT. For Pat2, days 1959 and 2323 mean sampling before the 11th and 12th RIT courses, and days 1962 and 2326 sampling after these courses, respectively. The values for healthy donors are shown as the sum of the examined values of all four persons.

CA yields and cytogenetic dose estimates based on unstable (dic+rc) and stable (tc+ti) CAs are presented in Table 2. For Pat1, the whole-body dose after one RIT course ranged between 0.52 and 0.95 Gy for solid staining and between 0.57 and 0.75 Gy for FISH. The respective doses of Pat2 ranged between 0.25 and 0.57 Gy and was 0.10 Gy for FISH (one-course conventional data were published in [16]).

Figure 1A shows the frequency and spectrum of CAs detected by mFISH in Pat1 on day 0. Among 1102 analyzed cells, 11 acentrics, 4 translocations, and one CCA were observed. These data were compared against the baseline CA frequencies detected by mFISH in four healthy donors. Figure 1B shows CAs detected by mFISH in Pat1 and Pat2 during RIT.

The frequency and spectrum of CAs induced by each RIT course (increment ∆) are presented in Figure 1C, which shows positive ∆-values only, despite a few negative increments observed. Since these increments were within the statistical errors, they are not shown in Figure 1C for clarity. Figure 2 shows the proportion of cells containing CA (solid symbols) and cells containing stable CA (open symbols) detected by mFISH before and after each RIT course.

## 3. Discussion

Following the RIT courses, both patients achieved complete remission, which was confirmed by the computer tomography (CT) scans. No abnormal accumulation of ^131^I in the body occurred. Several cases were reported when RIT was applied to a newborn ((3 × 1.48) GBq) [17] or children ((2 × 2.78) GBq) [18]. The treatment plan for Pat1 was milder ((3 × 0.55 + 0.62) GBq) and resulted in sustained remission and suppression of distant metastases in up to six years following RIT—a goal that is rarely fully achieved. None of the cases reported in [17,18] included a cytogenetic follow-up study of pediatric DTC patients. So, the data presented here are unique in that the RIT received by the DTC patient aged 1.6 was, for the first time, accompanied by the cytogenetic examination at all stages using three different assays, including the most advanced mFISH method.

The background level of all CAs classified by mFISH in Pat1 yielded 1.45 ± 0.36 CA/100 cells. The frequency averaged over healthy donors was 2.13 ± 0.42 CA/100 cells (Table 1). We used a different age group for comparison; nevertheless, the yields were not statistically different (*p* < 0.05). The background frequency of translocations in Pat1 was 0.36 CA/100 cells which exceeded the estimate of 0.05 recommended for such an age [19] using selective FISH painting. It should be noted that the latter estimate has a large statistical uncertainty since it was based on only three observations in children aged 1 to 4. Considering that sporadic CAs are viewed as a biological marker of cancer susceptibility [20], the increased CA level in Pat1 may be associated with an aggressive type of DTC. The presence of unstable CCAs (C/A/B–3/4/6) was unusual, too. A potential explanation could be related to the fact that the chest CT scan was performed 1.6 months before the examination. CT diagnostics in pediatric patients followed by an absorbed dose of about 10 mGy was shown to cause a detectable increase in CAs [21].

DTC patients who receive a very high total ^131^I activity represent another rare type. Pat2 had taken 10 courses of RIT in the period of 5.4 years before her first examination, at which time the total administered activity of ^131^I was 39 GBq. In the first examined sample (day 1959), among 11.8 CA/100 cells, 61.6% were translocations (Table 1), attributable to the previous exposures of PBL and bone marrow. The observed pattern was close to that described in [22], a 25-year follow-up study of a DTC patient, which reported 66 translocations in 669 cells, whereas we observed 54 translocations in 526 cells. No clonal aberrations, which reflect early damage to bone marrow stem cells and are defined as at least three cells with identical translocations, were found [23]. However, three variants of two cells with identical translocations were identified. Analysis of a larger number of cells is required to draw a final conclusion about this effect [24].

An increased proportion of cells containing CA (ccCA) and CA frequency was observed in both patients after each RIT course. Cells carrying unstable aberrations predominated (Figure 2). On day 0, the proportion of ccCA in Pat1 was 1.4%, of which 0.36% contained only stable CA (ccsCA). After the following RIT course, the ratio of ccCA to ccsCA was: (I)—(4.4/2.2)%; (II)—(6.2/2.1)%; (III)—(8.2/2.2)%; (IV)—(10.6/3.8)%. In Pat2, on the initial day (day 1959), the ccCA was 10.2%, of which 5.5% were ccsCA. The ratio of ccCA to ccsCA was: after 11th course—(16.4/7.6)%; after 12th course—(14.3/6.7)% (Figure 2). Thus, the ccsCA, which is associated with cancer risk and serves as a predictive biomarker of tumorigenesis [13], remained mostly unchanged over time in both patients. The decrease in ccCA between the RIT courses was associated with the elimination of aberrant PBL during the pool renewal. The drop was more pronounced in Pat2 since the interval between RIT courses was one year, while in Pat1, it was three months.

CCAs are known to be a marker of high-dose and/or high-LET exposure [13]. We found six CCAs in five out of eight samples in Pat1 (Table 1), while one CCA was found in the control group, which cannot be attributed to RIT. Only three out of six CCAs were stable (two insertions and a 3/3/3 translocation). Importantly, no CCAs were detected in the last two samples. In the follow-up examination of Pat1, neither the increase in the complexity of CCAs nor their accumulation was observed. In Pat2, CCAs of higher complexity were detected, although only a few of them were stable: 1 out of 5 (1/5), 0/6, 0/3, and 1/3 in samples one through four, respectively. Again, no accumulation of complex-type damage or increase in complexity was observed.

The assessment of the doses to healthy tissues based on unstable (dic+rc) and stable (tc+ti) CAs revealed that a one-time administration of RIT was safe, as the whole-body dose threshold of 2 Gy was not exceeded, with a statistical confidence of 95% (Table 2). The differences in doses using unstable and stable markers can be explained by extrapolation of the data obtained by selective FISH painting to the whole genome. This approach is known to result in certain misestimations [25,26].

The CA yields induced by a single RIT identified by mFISH in both patients do not exceed the CA yields induced by in vitro irradiation received at doses of (0.25–0.5) Gy (Figure 1C). Accounting for the dose rate effect [16], these CA frequencies correspond to a dose of about 40 percent larger, i.e., (0.35–0.7) Gy. Therefore, rough estimates of the accumulated dose as a result of the RIT courses using mFISH data are (1.4–2.8) Gy for Pat1 and (0.7–1.4) Gy for Pat2. Based on the analysis using conventional and FISH methods, the accumulated whole-body doses were found to be 2.9 and ~2.5 Gy for Pat1 and 0.82 and ~0.2 Gy for Pat2, respectively; these data are generally consistent with each other.

The frequency of tc+ti is recommended for retrospective biodosimetry [14]. Upon the completion of RIT, this value for Pat1 was 3.38 and 5.19 CA/100 GEcells using the FISH and mFISH assay, respectively. Similarly, for Pat2, it was 9.09 and 10.3 CA/100 GEcells (Table 2). As part of the model scenario of chronic exposure (G(T) = 0, α = 1.58 CA/100 cells, Gy^−1^ [16]), accounting for the age control [19], the accumulated dose is estimated at 2.1–3.2 Gy for Pat1 and 5.2–6.0 Gy for Pat2. This biodosimetric estimate is more accurate than the one based on the total administered activity, which is 0.55 and 11.1 Gy for Pat1 and Pat2, respectively (k_D_ = 0.238 mGy/MBq [5,16]).

The idea of using the mFISH capability to detect genome-wide CAs for biological dosimetry has been discussed elsewhere [26,27]. Dicentrics and translocations were initially used for biodosimetry. Recently, inversions and other complex exchanges were recommended for retrospective biodosimetry [28,29,30]. While evaluating RIT side effects, one study demonstrated a higher prognostic significance of acentrics induced in PBL by in vitro photon exposure compared to dicentrics [31]. A growing number of biodosimetric studies using mFISH revealed the need to include all visible CAs into consideration and develop a unified mFISH methodology to measure radiation damage with high accuracy for the purposes of biodosimetry and prediction of radiation tolerance [30]. We argue that mFISH should become a method of choice for biodosimetry—perhaps, not a universal one due to its costliness, but a standard for rare cases, such as the ones examined in this study.

## 4. Methods

This study was approved by the Ethics Committee of the A.F. Tsyb Medical Radiological Research Center, a branch of the National Medical Research Radiological Center of the Ministry of Health of the Russian Federation, according to the Helsinki declaration of 1975 (revised in 2013). All the participants were informed and signed consent for the study. The parents of the infantine patient were informed personally and signed a consent form.

The examined patients underwent a total thyroidectomy due to DTC followed by RIT at the MRRC (Obninsk) for thyroid remnants ablation and treatment of DTC’s distant metastases in lung and lymph nodes. Pat1 underwent four courses of RIT in a single year with a one-time administration of ^131^I in the range of (0.55–0.62) GBq or (0.055–0.054) GBq/kg, 2.29 GBq in total, all of which were followed by a cytogenetic examination. Pat2 underwent 12 courses of RIT during 6.4 years with a one-time administration of ^131^I in the range of (3.0–4.0) GBq or (0.034–0.045) GBq/kg, 46.7 GBq in total. The last two courses were followed by the cytogenetic examination. Blood samples were collected before and 3–4 days after each RIT course. For both patients, thyroid hormone withdrawal was performed four weeks before RIT. The thyroglobulin level (TG) in Pat1 on day 0 was 4112 ng/mL, which dropped dramatically and very quickly turned to 2.9 ng/mL on day 404. The TG in Pat2 on day 0 was 370 ng/mL, which also dropped dramatically and very quickly turned to 3.7 ng/mL on day 2323 (Figure 3). Neither patient had been previously treated with external beam radiotherapy, but five months before the RIT, Pat1 had undergone a chest CT scan.

Blood samples were also obtained from four healthy donors (two males, two females, aged 35–40, non-smokers), denoted as D1, D2, D3, and D4. These samples were used for in vitro γ-^60^Co exposure (dose rate = 0.82 Gy/min, linear energy transfer (LET) = 0.2 keV/µm) and analysis by mFISH (Table 1).

Culturing of the cells and chromosome preparation were carried out by the conventional method [14,15] at the MRRC (Obninsk). A total of 400 to 1068 cells per sample were analyzed by conventional analysis for unstable CAs—namely, dicentrics (dic), centric rings (rc), and excess acentric fragments (ace); additionally, 914 to 1001 cells per sample were analyzed by FISH painting of chromosomes 2, 4, and 12 (Metasystems, Germany) for stable CA translocations (tc+ti) (Table 1). The laboratory calibration curves for conventional and FISH assays were preliminarily refined by accounting for the dose rate effect [16].

The same samples were analyzed at JINR (Dubna) using a whole genome painting 24XCyte mFISH probe kit. CAs were identified and analyzed using ISIS/mFISH software (all MetaSystems, Germany), which assigns “pseudo colors” based on the unique combinations of fluorochromes, enabling the identification of all chromosomes (Figure 4A) and rearrangements between them. CAs were classified according to the mPAINT system [15]. CAs were subdivided into *simple breaks* (excess acentrics and truncated chromosomes) and *simple exchanges* originating from two breaks, which comprised translocations, dicenrics, centric, and acentric rings (including complete, incomplete, and one-way forms, as described in [15], (Figure 4B–F)). CAs containing three or more breaks in two or more chromosomes were classified as *complex aberrations* (CCAs, Figure 4B,D) and described by the ratio C/A/B—each letter representing the number of involved chromosomes, arms, and breaks, respectively. CAs were also classified according to their transmissibility to daughter cells: reciprocal translocations (complete form) were regarded as stable/transmissible; dicentrics, acentrics, and all non-reciprocal (incomplete and one-way) forms, which may lead to the loss of genetic material and cell death, as unstable/non-transmissible. CCAs were considered non-transmissible if they contained at least one non-transmissible part.

Biodosimetry using a linear-quadratic dose–response relationship of CA frequency was implemented [14,15]. The dose protraction factor, G(T), was calculated for each patient using the incomplete repair model and the period of isolated stay, T [16]. The data were summarized as mean ± SEM. SEM was calculated using the sampling distribution of CAs among cells if possible. In other cases, the Poisson distribution of CAs was assumed. The 95% CI of the absorbed doses was calculated [14].

## 5. Conclusions

The whole-body doses received by two rare DTC patients during one-time RIT courses determined using conventional and FISH methods did not exceed the safe threshold of 2 Gy. Cytogenetic biodosimetry implemented in the aftermath of RIT offers reliable radiation safety control, even if the patient is a toddler or if the total administered activity of ^131^I exceeds approximately 39 GBq.

The mFISH method is capable of determining genome-wide aberrations with high accuracy and identifying cells carrying stable CA with high precision to correctly assess the cytogenetic risk, i.e., the persistence of stable/transmissible heritable CAs in surviving cells, which are associated with cancer risk and serve as a predictive biomarker of tumorigenesis.

This study has demonstrated that successive RIT courses caused a sequential increase in the overall frequency of cells carrying CA mainly due to the induction of unstable CA, while the proportion of cells with stable CA remained mostly unchanged. Consequently, the cytogenetic risk was low for both examined patients. The obtained results offer a good outlook for RIT’s long-term implications. For certain rare cases, the results of cytogenetic examination could be used as an additional reliable substantiation to extend RIT.

## Figures and Tables

**Figure 1 ijms-24-05128-f001:**
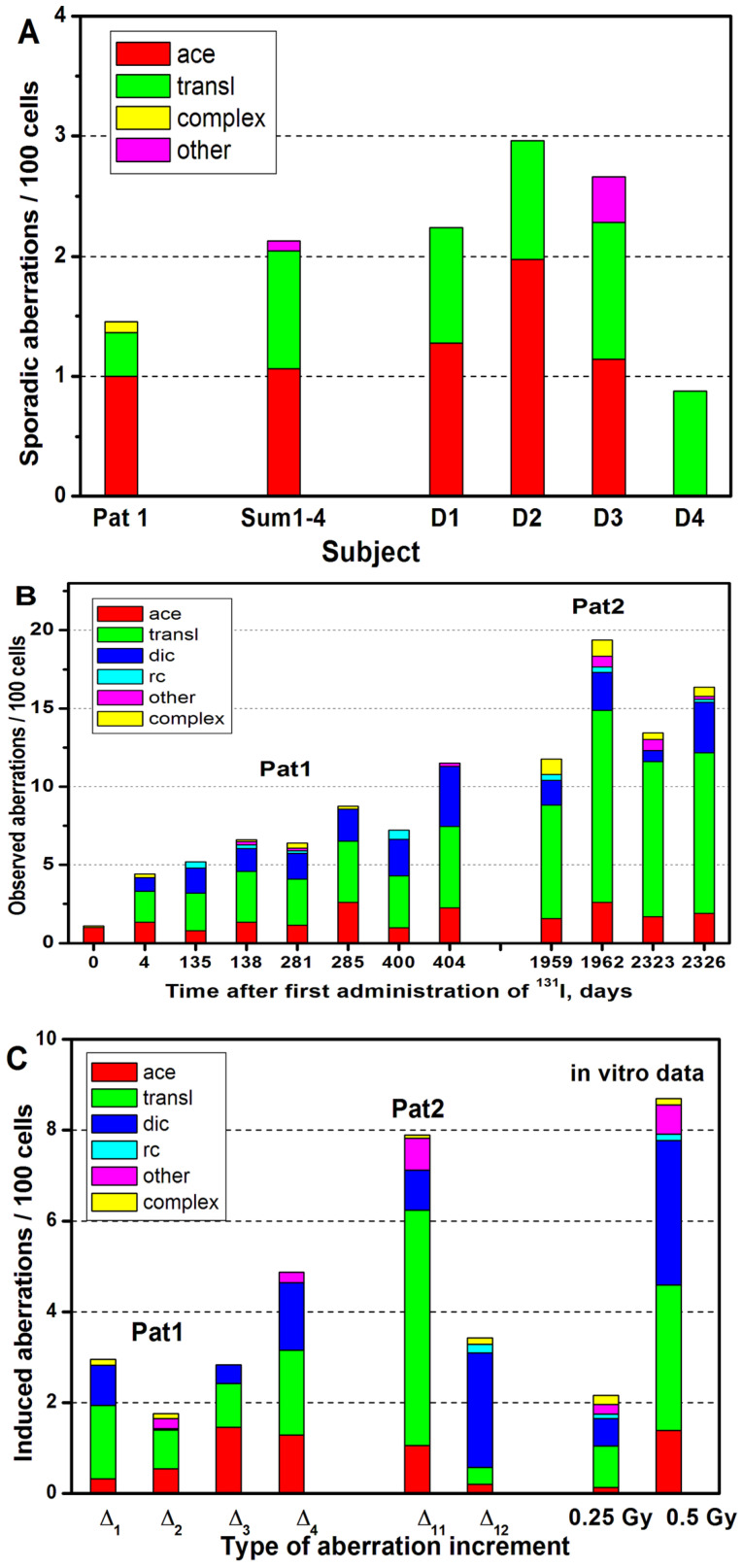
(**A**) The aggregate control frequency and spectrum of different types of chromosomal aberrations detected by mFISH in Patient 1 before RIT (day 0, 1102 cells) and in non-irradiated blood samples of four healthy donors: the sum for all four donors (Sum 1–4, 1223 cells) and separately (D1, D2, D3, D4); (**B**) the frequency and spectrum of different types of observed chromosome aberrations detected by mFISH as a result of 1–4 courses of RIT (Patient 1) and 11–12 courses of RIT (Patient 2); (**C**) the frequency and spectrum of different types of induced chromosomal aberrations in terms of increase, ∆, detected by the mFISH, as a result of 1–4 courses of RIT (Patient 1: ∆_1_–∆_4_) and 11–12 courses of RIT (Patient 2: ∆_11_–∆_12_). For comparison, the increases of induced aberrations (i.e., observed level minus control) are shown, amounting to four donors at 0.25 Gy (1003 cells) and 0.5 Gy (693 cells) obtained by in vitro γ-exposure of blood samples.

**Figure 2 ijms-24-05128-f002:**
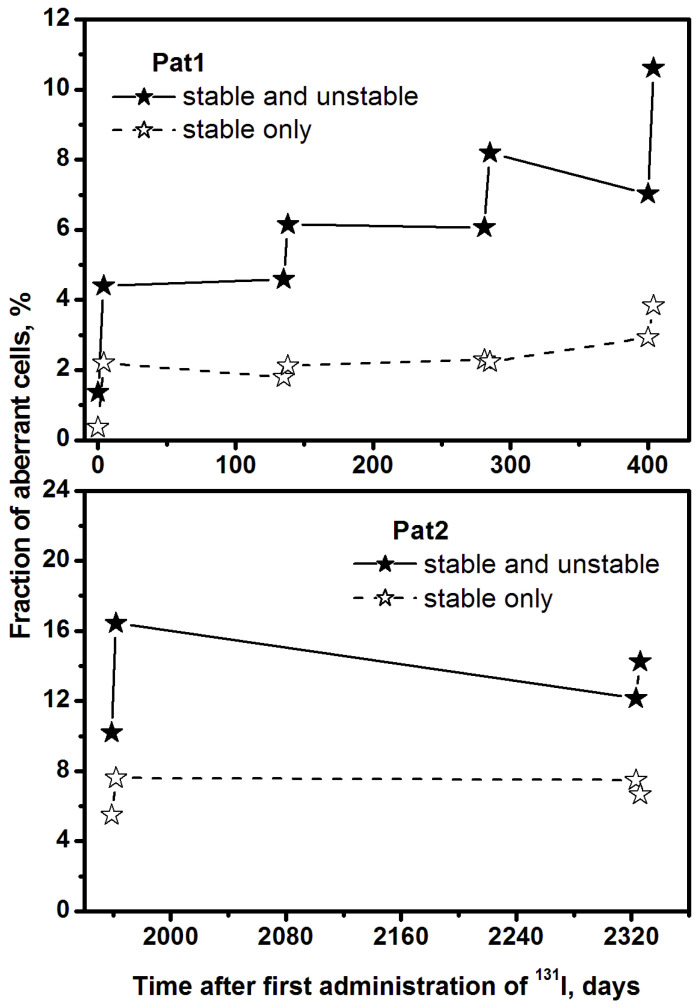
The fraction of aberrant cells detected by mFISH in examined patients. Asterisks indicate blood sampling before and after RIT (open symbols—percentage of stable aberrant cells; painted symbols—percentage of all aberrant cells).

**Figure 3 ijms-24-05128-f003:**
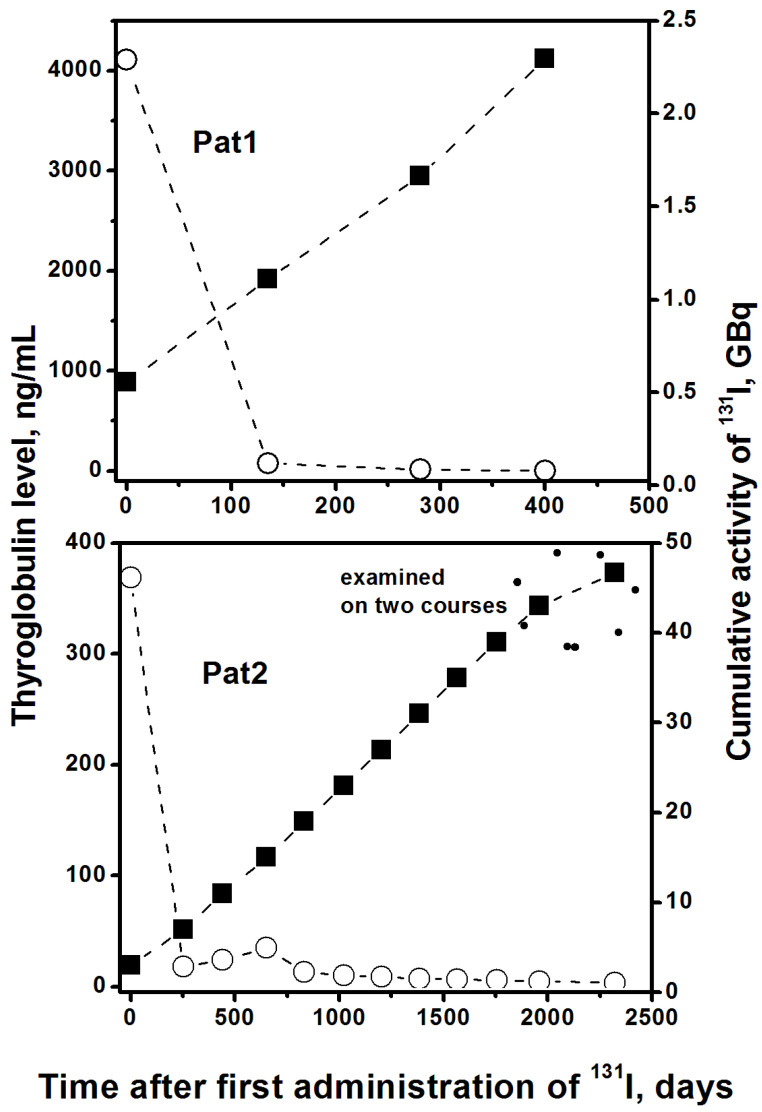
The cumulation of ^131^I drug administration (solid squares, right axis) for two examined patients, in comparison with the level of the oncomarker thyroglobulin (open circles, left axis) during the entire RIT, starting from the date of the first administration.

**Figure 4 ijms-24-05128-f004:**
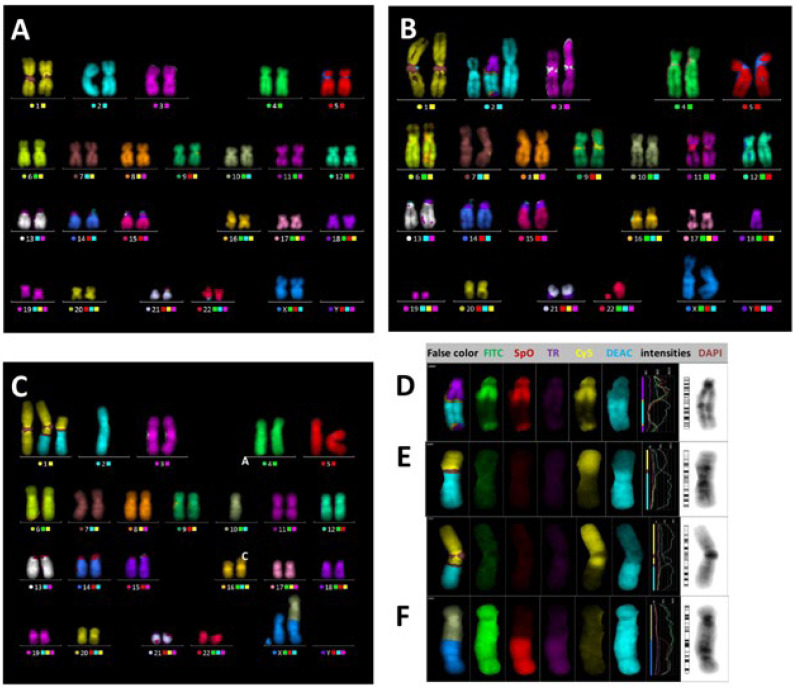
Patient 1, day 4. (**A**) Normal human female karyogram obtained by the mFISH method. Each chromosome pair is identified by software on the basis of a unique combination of fluorescent dyes (the squares to the right from the chromosome number) and is presented in pseudo colors generated by ISIS/mFISH software for analysis convenience. (**B**) Complex aberration tr 18′-2-18 + T 2′ described by the ratio C/A/B 2/2/3. (**D**) Insertion tr 18′-2-18. Single color gallery provides full information about painted items (from left to the right): pseudo colors, five fluorochromes, intensity profiles of all fluorochromes, and DAPI image. Patient 2, day 2323. (**C**) Two simple exchanges were observed: reciprocal translocation tr 1′-2 + tr 2′-1—stable aberration (**E**) dicentric 10′-X’ + ace 10-X—unstable aberration (**F**). The cell containing at least one unstable aberration is regarded as unstable.

**Table 1 ijms-24-05128-t001:** Chromosome aberrations in blood lymphocytes of the DTC patients detected by different methods before and after RIT and in healthy donors.

Patient 1								
	Day 0	Day 4	Day 135	Day 138	Day 281	Day 285	Day 400	Day 404
	Conventional analysis using Giemsa painting							
Number of cells:	1068	1000	500	550	500	400	500	500
aberrant cells	3	33	14	42	16	26	29	47
acentrics	1	20	8	27	16	16	16	20
centric rings	0	5	0	4	0	5	3	6
dicentrics	3	12	6	15	4	8	11	26
	FISH analysis using selective painting of chromosomes 2, 4, 12							
Number of cells:	-	-	1000	1000	1000	1000	1000	1000
reciprocal translocations (tc)	-	-	8	11	5	8	5	8
non-reciprocal translocations (ti)	-	-	2	3	5	6	1	4
deletions	-	-	5	9	3	7	4	6
	mFISH analysis using whole genome painting							
Number of cells:	1102	454	501	893	610	537	512	443
aberrant cells	15	20	23	55	37	44	36	47
stable aberrant cells	4	10	9	19	14	12	15	17
reciprocal translocations	4	9	10	22	14	15	16	18
non-reciprocal translocations	0	0	2	7	4	6	1	5
acentrics	11	6	4	12	7	14	5	10
centric rings	0	0	1	2	1	0	3	0
dicentrics	0	4	8	13	10	11	12	17
other simple exchanges *	0	0	0	2	1	0	0	1
complex aberrations	1	1	0	1	2	1	0	0
Total breaks	25	35	46	107	74	83	71	92
**Patient 2**								
	Day 1959		Day 1962		Day 2323		Day 2326	
	Conventional analysis using Giemsa painting							
Number of cells:	500 **		514 **		657		500	
aberrant cells	18		43		51		31	
acentrics	8		28		33		14	
centric rings	5		4		4		3	
dicentrics	10		20		20		18	
	FISH analysis using selective painting of chromosomes 2, 4, 12							
Number of cells:			-		1000		914	
reciprocal translocations	-		-		16		16	
non-reciprocal translocations	-		-		12		10	
deletions	-		-		1		3	
	mFISH analysis using whole genome painting							
Number of cells:	510		578		707		526	
aberrant cells	52		95		86		75	
stable aberrant cells	28		44		53		35	
reciprocal translocations	34		57		60		44	
non-reciprocal translocations	3		14		10		10	
acentrics	8		15		12		10	
centric rings	2		2		0		1	
dicentrics	8		14		5		17	
other simple exchanges *	0		4		5		1	
complex aberrations	5		6		3		3	
Total breaks	118		219		183		166	
Aberrations detected by mFISH in healthy donors induced by ^60^Co γ-irradiation								
			Dose, Gy					
			0		0.25		0.5	
Number of cells:			1223		1003		693	
aberrant cells			26		43		69	
stable aberrant cells			10		15		23	
reciprocal translocations			10		15		25	
non-reciprocal translocations			2		4		4	
acentrics			13		12		17	
centric rings			0		1		1	
dicentrics			0		6		22	
other simple exchanges *			1		3		5	
complex aberrations			0		2		1	

* acentric rings, inversions; ** data were published partly [16].

**Table 2 ijms-24-05128-t002:** The aberration yield and cytogenetic dose estimate for the DTC patients using different cytogenetic assays.

Patient 1								
Conventional analysis of (dic+rc) using Giemsa painting								
Days	Pre-treatment		T, h	Post-treatment		Increment ∆M ± SEM *	*G*(T)	Dose, Gy (CI **)
	cells	M_1_ ± SEM *		cells	M_2_ ± SEM *			
0–4	1068	0.28 ± 0.21	92	1000	1.70 ± 0.41	1.42 ± 0.62	0.246	0.52 (0–0.84)
135–138	500	1.20 ± 0.49	68	550	3.45 ± 0.78	2.25 ± 1.27	0.251	0.70 (0–1.19)
281–285	500	0.80 ± 0.40	92	400	3.25 ± 0.89	2.45 ± 1.29	0.246	0.74 (0–1.23)
400–404	500	2.80 ± 0.74	92	500	6.40 ± 1.13	3.60 ± 1.87	0.246	0.95 (0–1.53)
FISH analysis of (tc+ti) using selective painting of chromosomes 2, 4, 12								
Days	Pre-treatment		T, h	Post-treatment		Increment ∆F ± SEM *	*G*(T)	Dose, Gy (CI **)
	cells	F_1_ ± SEM *		cells	F_2_ ± SEM **			
135–138	1000	3.19 ± 1.01	68	1000	4.47 ± 1.20	1.28 ± 2.21	0.251	0.57 (0–1.48)
281–285	1000	3.19 ± 1.01	92	1000	4.47 ± 1.20	1.28 ± 2.21	0.246	0.57 (0–1.49)
400–404	1000	1.92 ± 0.78	92	1000	3.83 ± 1.11	1.92 ± 1.87	0.246	0.75 (0–1.49)
**Patient 2**								
Conventional analysis of (dic+rc) using Giemsa painting								
Days	Pre-treatment		T, h	Post-treatment		Increment ∆M ± SEM *	*G*(T)	Dose, Gy (CI **)
	cells	M_1_ ± SEM *		cells	M_2_ ± SEM *			
1959–1962	500	3.00 ± 0.81	68	514	4.67 ± 1.01	1.67 ± 1.82	0.251	0.57 (0–1.23)
2323–2326	657	3.65 ± 0.85	68	500	4.20 ± 0.94	0.55 ± 1.79	0.251	0.25 (0–1.05)
FISH analysis of (tc+ti) using selective painting of chromosomes 2, 4, 12								
Days	Pre-treatment		T, h	Post-treatment		Increment ∆F ± SEM **	*G*(T)	Dose, Gy (CI **)
	cells	F_1_ ± SEM **		cells	F_2_ ± SEM **			
2323–2326	1000	8.95 ± 1.69	68	914	9.09 ± 1.78	0.14 ± 3.47	0.251	0.10 (0–1.64)

* standard error of the mean; ** 95% confidence interval.

## Data Availability

The datasets used and/or analyzed during the current study are available from the corresponding author upon reasonable request.

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
