# Peer review of "Cytogenetic Damage Induced by Radioiodine Therapy: A Follow-Up Case Study"

_ijms, 2023, doi:10.3390/ijms24065128_

Round 1
Reviewer 1 Report
The manuscript entitled „Cytogenetic damage induced by radioiodine therapy: a follow-up case study” by Khvostunov et al. is of interest for the Journal’s readers, being also rich in experimental data, which are properly analysed.
However, the authors are asked to explain what novelty their study presents, as radioiodine therapy is known, including dose threshold of 2 Gy. I do not see what is new. Nevertheless, on sudying this manuscript, I wonder about a possible relationship between glutathione in blood and other fluids of the body and the presence of radioactive iodine. Some papers reported a so-called anti-toxic role of glutathione and cysteine-containing peptide (Popa et al. Isotopes in Environmental and Health Studies, 2, 105-116, 2007; Murariu et al., Rev. Roum. Chim., 54(8) 741-747, 2009; Popa et al., Protective role of cysteine-based peptides against the radiotoxic cations within several germination experiments. In Metal Ions in Biology and Medicine: XVIII Nutrition, vol. 10. Eds. Ph. Collery, I. Maymard, T. Theophanides, L. Khassanova, T. Collery, John Libbey Eurotext, Paris, 2008, pp. 838-842; or Murariu et al. Contributions to the state of the art in radionuclides–plants interaction field …. Springer, 91-106, 2014).
Although the manuscript is rich in experimental data, changes in glutathione levels before and after treatment with iodine-131 may suggest its role in improving health. In addition, increased glutathione levels may protect the body against radionuclides, so the dose of radioactive iodine could be raised. However, these questions might be suggestions for another article.
Minor points.
We suggest using the model paper (template) recommended by the journal. The whole manuscript should thus re-formate. In such case, the authors may improve much the qualiy of their article. We also consider that the authors should inroduce references according to the journal format.
Author Response
Response to Reviewer 1 Comments
Point 1: However, the authors are asked to explain what novelty their study presents, as radioiodine therapy is known, including dose threshold of 2 Gy. I do not see what is new. Nevertheless, on sudying this manuscript, I wonder about a possible relationship between glutathione in blood and other fluids of the body and the presence of radioactive iodine. Some papers reported a so-called anti-toxic role of glutathione and cysteine-containing peptide (Popa et al. Isotopes in Environmental and Health Studies, 2, 105-116, 2007; Murariu et al., Rev. Roum. Chim., 54(8) 741-747, 2009; Popa et al., Protective role of cysteine-based peptides against the radiotoxic cations within several germination experiments. In Metal Ions in Biology and Medicine: XVIII Nutrition, vol. 10. Eds. Ph. Collery, I. Maymard, T. Theophanides, L. Khassanova, T. Collery, John Libbey Eurotext, Paris, 2008, pp. 838-842; or Murariu et al. Contributions to the state of the art in radionuclides–plants interaction field …. Springer, 91-106, 2014).
Although the manuscript is rich in experimental data, changes in glutathione levels before and after treatment with iodine-131 may suggest its role in improving health. In addition, increased glutathione levels may protect the body against radionuclides, so the dose of radioactive iodine could be raised. However, these questions might be suggestions for another article.
Response 1:
In this article, the dose threshold of 2 Gy is used only for comparison with the available results. The novelty of the work represents the priority study of the consequences of radioiodine therapy for the infant using the mFISH method throughout the course of four administrations. None of the cases reported early included a cytogenetic follow-up study of pediatric DTC patients.
The changes in glutathione levels before and after treatment with iodine-131 could definitely be an important factor influencing the effects of radioiodine therapy. In partucular the increased glutathione level may protect the patient body against side effects of radionuclides and provide an opportunity to increase the therapeutic effect. All of this could be the subject of future research.
Point 2:. We suggest using the model paper (template) recommended by the journal. The whole manuscript should thus re-formate. In such case, the authors may improve much the qualiy of their article. We also consider that the authors should inroduce references according to the journal format.
Response 2:
The whole manuscript was re-formated using template recommended by the journal including sequence and list of sections, abstract format, font type, references etc.

Reviewer 2 Report
The manuscript by Khvostunov et al reports a case-study investigation of chromosomal aberrations in peripheral blood lymphocytes induced by radio iodine therapy of rare thyroid cancer. The study is relevant, well-performed and is of interest for researchers and clinicians working in this area. The overall merit of this report is high, but a few issues could be improved to make it even better.
1. It is not entirely clear if peripheral blood lymphocytes have been isolated or analyzed directly in blood smears. 1a (related to #1). Do the data obtained allow for the analysis of chromosomal aberrations in specific populations of PBL. IF this is possible, it would be of interest.
2. The authors state (p. 4) that TG dropped to a certain level at the time of the last treatment in the series. This statement is correct, but creates the impression that this decrease is gradual. But it would be more relevant to indicate that TG dropped dramatically and very fast.
3. It would be helpful to the readers, if the dose per kg was indicated for both patients.
4. p. 7 "ref. Table I" is probably a typo.
5. p. 3 The statement goes that all participants were informed and signed a consent form. For one of them it was probably done by parents or guardians.
Author Response
Response to Reviewer 2 Comments
Point 1: It is not entirely clear if peripheral blood lymphocytes have been isolated or analyzed directly in blood smears. 1a (related to #1). Do the data obtained allow for the analysis of chromosomal aberrations in specific populations of PBL. IF this is possible, it would be of interest.
Response 1: Peripheral blood lymphocytes were analyzed as a part of whole blood sampled from patient vien. All the lymphocytes in the blood vessels are at rest, i.e. in the stage G0, therefore they must be stimulated by special chemicals, namely, PHA here to divide in order to analyze induced chromosomal aberrations on mitotic chromosomes. According to IAEA mannual 2011 the most specific sub-populations of PBL that can be stimulated to divide by PHA are T- lymphocytes. So, this specific population of PBL is used for biological dosimetry as a “gold standard” for many years now (IAEA, 2011).
Point 2:. The authors state (p. 4) that TG dropped to a certain level at the time of the last treatment in the series. This statement is correct, but creates the impression that this decrease is gradual. But it would be more relevant to indicate that TG dropped dramatically and very fast.
Response 2: The statement that TG dropped adjusted to reflect that TG dropped dramatically and very fast.
Point 3: It would be helpful to the readers, if the dose per kg was indicated for both patients.
Response 3: The administred activities per kg were indicated for both patients.
Point 4: p. 7 "ref. Table I" is probably a typo.
Response 4: The typo was fixed.
Point 5: p. 3 The statement goes that all participants were informed and signed a consent form. For one of them it was probably done by parents or guardians
Response 5: The existence of informational consent from the parents of the infantine patient was added to the text of the article.
